# Probing the Active Site of Class 3 L-Asparaginase by Mutagenesis: Mutations of the Ser-Lys Tandems of ReAV

**DOI:** 10.3390/biom15070944

**Published:** 2025-06-29

**Authors:** Kinga Pokrywka, Marta Grzechowiak, Joanna Sliwiak, Paulina Worsztynowicz, Joanna I. Loch, Milosz Ruszkowski, Miroslaw Gilski, Mariusz Jaskolski

**Affiliations:** 1Institute of Bioorganic Chemistry, Polish Academy of Sciences, Noskowskiego 12/14, 61-704 Poznan, Poland; kpokrywka@man.poznan.pl (K.P.); mgrzech@ibch.poznan.pl (M.G.); sliwiak@man.poznan.pl (J.S.); worsztynowiczpaulina@gmail.com (P.W.); mruszkowski@ibch.poznan.pl (M.R.); mirek@amu.edu.pl (M.G.); 2Department of Biotechnology and Food Microbiology, Poznan University of Life Science, Wojska Polskiego 48, 60-627 Poznan, Poland; 3Department of Crystal Chemistry and Crystal Physics, Faculty of Chemistry, Jagiellonian University, Gronostajowa 2, 30-387 Krakow, Poland; joanna.loch@uj.edu.pl; 4Department of Crystallography, Faculty of Chemistry, Adam Mickiewicz University, Uniwersytetu Poznanskiego 8, 61-614 Poznan, Poland

**Keywords:** hydrolase, amidohydrolase, L-asparaginase, leukemia, metalloprotein, site-directed mutagenesis, Nessler reaction, ITC, X-ray crystallography

## Abstract

The ReAV enzyme from *Rhizobium etli*, a representative of Class 3 L-asparaginases, is sequentially and structurally different from other known L-asparaginases. This distinctiveness makes ReAV a candidate for novel antileukemic therapies. ReAV is a homodimeric protein, with each subunit containing a highly specific zinc-binding site created by two cysteines, a lysine, and a water molecule. Two Ser-Lys tandems (Ser48-Lys51, Ser80-Lys263) are located in the close proximity of the metal binding site, with Ser48 hypothesized to be the catalytic nucleophile. To further investigate the catalytic process of ReAV, site-directed mutagenesis was employed to introduce alanine substitutions at residues from the Ser-Lys tandems and at Arg47, located near the Ser48-Lys51 tandem. These mutational studies, along with enzymatic assays and X-ray structure determinations, demonstrated that substitution of each of these highly conserved residues abolished the catalytic activity, confirming their essential role in enzyme mechanism.

## 1. Introduction

Asparaginases (EC 3.5.1.1) catalyze the hydrolysis of L-asparagine, generating L-aspartate and ammonia. They are grouped into three completely unrelated structural classes that are further subdivided into five types [1,2]. Class 1 L-asparaginases, formerly known as bacterial asparaginases, are homotetrameric enzymes of two types, with millimolar (type I, e.g., EcAI) and micromolar (type II, e.g., EcAII) substrate affinities [3,4,5,6]. L-asparaginases from Class 2 (plant or type III asparaginases) belong to the family of N-terminal nucleophile (Ntn) amidohydrolases that are initially produced as inactive precursors and acquire catalytic activity after autoproteolytic cleavage [7,8,9]. They are (αβ)_2_ heterotetramers composed of two α and two β subunits [10] and have a rather low, millimolar substrate affinity [11,12]. L-asparaginases of Class 3, first identified in the symbiotic nitrogen-fixing bacterium *Rhizobium etli*, are represented by two isoforms (types IV and V) differentiated by their thermostability, electrophoretic mobility, and expression profiles. The constitutive ReAIV enzyme (*R. etli* type IV) is thermostable, whereas the ReAV enzyme (*R. etli* type V) is inducible and thermolabile [13,14]. The sequence homology between ReAIV and ReAV is relatively low, with only about 30% identity. However, despite low sequence identity, the two isoforms exhibit a high degree of structural similarity, with an almost identical dimer architecture and active site [15,16]. Their substrate affinities are in the low millimolar range (K_M_ 1.5 mM for ReAIV and 2.1 mM for ReAV), and they exhibit complementary biochemical properties [17].

The high-affinity type II bacterial L-asparaginases have been used to treat acute lymphoblastic leukemia (ALL) for many years, following their identification as potential antileukemic agents in 1961 [18]. The L-asparagine-dependent leukemic cells are starved to death upon L-asparaginase administration, which efficiently removes this amino acid from circulation [19,20,21]. Currently, Class 1 type II L-asparaginases from *Escherichia coli* and *Erwinia chrysanthemi* (now known as *Dickeya dadantii*) are highly effective antileukemic drugs due to their low K_M_ values; however, they elicit serious side effects, including hypersensitivity, neurotoxicity, and pancreatitis [22,23]. In view of the micromolar (~50–70 µM) concentration of L-asparagine in circulation [24,25], the millimolar affinity of Class 2 L-asparaginases is too low for use in ALL therapy. Alternative sources of antileukemic L-asparaginases are, therefore, urgently needed. The recently solved crystal structure of the Class 3 L-asparaginase ReAV revealed an enzyme that bears no similarity to Class 1 or Class 2 proteins [15]. This suggests that ReAV likely operates via a distinct catalytic mechanism, making it an interesting candidate for the development of novel antileukemics. Apart from medicinal applications, L-asparaginases are also important in the food industry, where they are used for enzymatic removal of free L-asparagine, thus preventing acrylamide formation during high-temperature processing of fried and baked foods [26,27].

ReAV is a homodimeric protein with an α/β protomeric fold [15] (Figure 1A). The active site is marked by a zinc cation, coordinated by Cys135, Cys189, and, unusual in this role, Lys138 (Figure 1B). It has been shown that Zn^2+^ binding is highly selective [15] and that the metal center may play a role in substrate recognition [17,28]. Close to the zinc binding site, two Ser-Lys tandems (Ser48-Lys51, Ser80-Lys263) are located, with the hydroxyl group of Ser48 typically surrounded by three electron-density peaks, interpreted as a tightly H-bonded water triad. Extensive analyses of over 45,000 bacterial genomes, combined with comparisons with available structural homologs, led to the identification of two highly conserved patterns in the ReAV sequence [29]. Residues from Pattern I include Cys135, Lys138, and Cys189, directly involved in zinc coordination, as well as residues located near the metal binding site (His139, Tyr156, Asp187, and Cys249). Pattern II consists of residues forming the two Ser-Lys tandems (Ser48-Lys51, Ser80-Lys263) and the Arg47 residue located in their neighborhood. We recently published a report on site-directed mutagenesis of Pattern I, which includes a comprehensive analysis of the crystal structures of the generated ReAV mutants combined with kinetic studies [28]. The focus of the present paper is on mutagenesis of Pattern II residues.

The putative active site of ReAV in the area comprising the two Ser-Lys tandems resembles the active site of the homologous *E. coli* serine β-lactamase (PDB ID 1fqg), where Ser70, corresponding to Ser48 in ReAV, acts as the catalytic nucleophile, from which the proton is transferred to the uncharged side chain amine group of Lys73 (Lys51 in ReAV), acting as a general base [30] (Figure 1C,D). Nonetheless, the identity of the general base responsible for activating this serine residue remains a matter of debate. While one hypothesis attributes this role to Lys73, an alternative proposal implicates the conserved Glu166 [31,32]. The strong hydrogen bond between the second Ser-Lys pair, namely Ser130 and Lys234 (Ser80 and Lys263 in ReAV), promotes the second proton transfer due to the decreased proton affinity of the serine 80 hydroxyl group [30]. Lys234 has also been postulated to stabilize the transition state of the catalytic reaction [33].

The unusual hydration pattern of Ser48, observed in several ReAV crystal structures reported by Loch et al. (2021) [15] and by Pokrywka et al. (2024) [28], strongly suggests that this residue is unique in the protein structure and thus a good candidate as the catalytic nucleophile. This prediction was confirmed by the crystal structures of WT ReAV and its mutants in complex with the L-Asn substrate, which additionally showed that the metal cation serves as an anchor for the substrate α-amino and α-carboxyl groups [34]. Taken together, the previous results support a double-displacement mechanism, with catalysis proceeding through a covalent β-acyl-enzyme intermediate (Figure 1). However, the current crystallographic data do not fully resolve the catalytic scenario, as details regarding the second nucleophilic attack, i.e., hydrolysis of the ester intermediate, are lacking. In this work, we generated five alanine mutants of ReAV, including mutation of the Ser48 nucleophile, mutations of the remaining residues from the Ser-Lys tandem involved in proton shuttle (i.e., Lys51, Ser80, and Lys263), and the mutation of the conserved Arg47 residue located in the close vicinity of the tandems and assisting in the correct positioning of the substrate molecule [34]. The main goal of the present work is to elucidate the structural and catalytic roles of the selected residues, with a view to better understand the catalytic mechanism of Class 3 L-asparaginases.

## 2. Methods

### 2.1. Site-Directed Mutagenesis

Site-directed mutagenesis of ReAV was performed using two methods: the polymerase incomplete primer extension (PIPE) technique [35] for the R47A and S48A mutants, and the Q5-site mutagenesis protocol for the K51A, S80A, and K263A variants. Alanine substitutions were introduced at designated positions using specific primers, as listed in Table 1. The pET151-D-ReAV plasmid, which contains the original protein sequence, served as a template for Polymerase Chain Reaction (PCR) to amplify the target sequence. To minimize transformation background, template DNA was digested with the DpnI restriction enzyme, and the resulting reaction products were used to transform *E. coli* BL21-Gold (DE3) competent cells (Agilent Technologies). After overnight incubation on LB agar plates supplemented with 100 µg·mL^−1^ ampicillin, single colonies were inoculated into 4 mL of LB medium with ampicillin and grown overnight at 37 °C. Following incubation, plasmids were isolated and subjected to Sanger sequencing (Genomed, Poland) to verify the presence of the desired mutations.

### 2.2. Protein Expression and Purification

Protein expression and purification followed the protocol established by Pokrywka et al. (2024) [28]. Following transformation, a glycerol stock was used to inoculate 10 mL of LB medium supplemented with 100 µg·mL^−1^ ampicillin. After overnight incubation at 37 °C, the preculture was transferred to 1 L of LB medium with ampicillin and cultured at 37 °C until the OD_600_ reached ~0.7. The temperature was then lowered to 18 °C, and protein expression was induced by adding 0.2 mM isopropyl β-D-thiogalactopyranoside (IPTG). After overnight induction, cells were harvested by centrifugation at 6150× *g* for 10 min at 4 °C, and the cell pellet was re-suspended in 35 mL of binding buffer [50 mM Tris-HCl pH 8.0, 500 mM NaCl, 20 mM imidazole, 10% *v*/*v* glycerol, 1 mM tris(2-carboxyethyl)phosphine (TCEP)]. The cell suspension was frozen and stored at −80 °C. After thawing, cells were lysed by sonication, and the lysate was clarified by centrifugation at 31,500× *g* for 30 min at 4 °C. The supernatant was loaded onto an affinity column filled with HisTrap HP resin and equilibrated with binding buffer. The protein was eluted using elution buffer (50 mM Tris-HCl pH 8.0, 500 mM NaCl, 300 mM imidazole, 10% *v*/*v* glycerol, 1 mM TCEP), and the eluate was dialyzed overnight with Tobacco Etch Virus (TEV) protease at 4 °C against 50 mM Tris-HCl buffer pH 8.0, containing 500 mM NaCl and 1 mM TCEP. After TEV protease digestion, the sample was passed again through the HisTrap column to remove His-tag debris and the His-tagged TEV protease. The flow-through containing the recombinant ReAV protein was collected. The protein was concentrated and loaded on a HiLoad 16/60 Superdex 200 column equilibrated with 50 mM Tris-HCl buffer pH 8.0, containing 150 mM NaCl and 1 mM TCEP. Protein fractions were analyzed for purity by SDS-PAGE, and those containing pure protein were pooled and concentrated using 30,000 MWCO Amicon Ultra-15 centrifuge filters.

### 2.3. Measurements of Enzymatic Activity of ReAV Mutants

Asparaginase activity of the mutants (R47A, S48A, K51A, S80A, K263A) was measured by the Nesslerization reaction [36] as previously described by Sliwiak et al. (2024) [17] with the following settings: 10 mM L-Asn and 10 µM enzyme in 10 mM Tris-HCl pH 9.0 buffer incubated for 10–30 min at room temperature. Additionally, ITC-MIM (Isothermal Titration Calorimetry—Multiple Injection Method) [37] kinetic measurements with MicroCal iTC200 or PEAQ-ITC calorimeters (Malvern) were performed in 10 mM Tris-HCl pH 9.0 buffer, as described in Sliwiak et al. (2024) [17]. Briefly, heat-rate shift experiments were performed by injecting L-Asn at 100 mM concentration (in the syringe) in 20 aliquots of 1.8 µL with 60 s intervals (PEAQ-ITC), or 12 aliquots of 3 µL with 80 s intervals (iTC-200), into the reaction cell with the enzyme kept at 10–20 µM concentration. All ITC measurements were taken at 37 °C, with stirring at 700 rpm and differential power set to 10 μcal·s^−1^, and were conducted in technical triplicates. For some muteins (R47A, S48A, K51A), the measurements were biological duplicates or triplicates, i.e., were performed for different protein samples expressed and purified from bacteria after separate transformation procedures.

### 2.4. Crystallization

Before crystallization, the protein solutions were concentrated to 15–21 mg·mL^−1^ and supplemented with *n*-dodecyl-β-D-maltoside (DDM) to a final concentration of 0.5–1% *v*/*v*. Crystallization screening was performed using vapor diffusion in the hanging drop setup, following the conditions established for ReAV muteins by Pokrywka et al. (2024) [28]. Crystals were grown at 19 °C over a period of 2–5 days, using the crystallization solutions listed in Table 2. Crystals were cryoprotected in their mother liquor supplemented with ethylene glycol, vitrified in liquid nitrogen, and stored until X-ray data collection.

### 2.5. X-Ray Data Collection, Crystal Structure Solution, and Refinement

X-ray diffraction data were collected at the P13 EMBL beamline of the Petra III synchrotron at DESY, Hamburg, Germany. The diffraction images were processed using XDS [38]. The crystal structures of the ReAV mutants were solved by molecular replacement using Phaser [39] with the WT ReAV structure PDB ID 7os5 as the starting model. The structures were refined with Phenix [40] using anisotropic or TLS protocols. The electron density maps and solvent molecules were inspected in Coot [41]. All crystal structures were standardized in the crystallographic unit cell using the ACHESYM server [42]. Data collection and structure refinement statistics are summarized in Table 3. Structural illustrations were generated with PyMOL [43] and annotated in BioRender (https://www.biorender.com accessed on 1 May 2025).

## 3. Results and Discussion

### 3.1. Exploring the Importance and Properties of Selected ReAV Residues

Five conspicuous residues, namely Arg47, Ser48, Lys51, Ser80, and Lys263, are located in the close vicinity of the metal binding site of ReAV (Figure 1B). The positively charged Arg47 side chain participates in salt-bridge interactions with the carboxylate side chains of Asp187, Asp267, and Glu17 from the adjacent protein chain, creating a network of H-bonds that supports the dimer structure. Furthermore, the crystal structure of the ReAV protein in complex with L-Asn showed that the guanidinium group of Arg47 stabilizes the position of the substrate α-carboxylate group [34]. A pattern of three closely spaced electron-density peaks clustered around the hydroxyl group of Ser48, interpreted as water molecules forming exceptionally strong hydrogen bonds with the Oγ atom of Ser48 (O…O distances 2.20–2.40 Å) and within their triangular arrangement (O…O 2.20–2.30 Å), highlighted the unique nature of this residue. Ser48 was hypothesized to be the primary nucleophile, and the predictions have been corroborated by the crystal structures of WT ReAV and its two mutants (K138A and K138H) in complex with the L-Asn ligand [34]. Moreover, the side chain of Ser48 interacts with the Oγ atom of Ser80, and this connection stabilizes the protein structure and forces the Ala79-Ser80 peptide into a non-planar configuration (Cα-C-N-Cα torsion angle ~150°) [15]. Near these Ser residues, there are two Lys residues, Lys51 and Lys263, respectively. The Nζ atom of Lys51 forms H-bonds to the Oγ atoms of Ser48 and Ser80, as well as to the carbonyl O atom of Ala79. Lys51 plays the role of the nucleophilic activator, acting as a base for proton abstraction from Ser48 [34]. The side chain of Lys263 is hydrogen-bonded to the Oγ atom of Ser80, carbonyl oxygen of Leu264, and a water molecule. Lys263, together with Ser80 and Lys51, is involved in a proton shuttling network during the asparaginase reaction. The proposed mechanism of ReAV is shown in Figure 1.

### 3.2. Expression and Purification of the Generated ReAV Mutants

The five carefully selected residues were subjected to site-directed mutagenesis and substituted with alanine. DNA sequencing confirmed the correctness of the mutant sequences. However, it must be noted that inspection of the electron density maps of the K51A mutant revealed the presence of an unintended substitution of His238 with tyrosine. The fact that experimental electron density maps were able to unambiguously reveal an unexpected amino acid substitution, to be later confirmed by plasmid sequencing, underscores the power of the crystallographic method in structural biology, as discussed in more detail in Section 3.4.3.

The alanine mutants of Arg47, Ser48 and Lys263 were expressed in *E. coli* in substantial amounts in soluble form (~50 mg·L^−1^ of bacterial culture). A significantly higher expression yield (~80 mg·L^−1^) was obtained for the K51A variant. The alanine substitution of Ser80 resulted in the formation of highly aggregated protein in inclusion bodies; however, approximately 10 mg·L^−1^ of the protein could be recovered from the cell lysate.

### 3.3. Enzymatic Activity and Kinetic Measurements

Within the detection limits of the methods used, even at high enzyme concentrations intentionally used to increase the expected signal, neither the Nesslerization reaction detected ammonia production by the studied muteins, nor did the ITC kinetic assays reveal any exothermic effect resulting from asparagine hydrolysis (Appendix A, Appendix A). The observed signals were below the detection limits of 0.01 U for the Nessler method [44] and 0.1 μcal·s^−1^ for ITC [45]. These results support the hypothesis that each of the studied residues is essential for catalysis.

### 3.4. Description of the Crystal Structures of the Mutant Proteins

All five ReAV mutants were successfully crystallized. Single crystals with thin-plate morphology were obtained at different crystallization conditions; however, the crystal quality varied depending on the introduced mutation. The investigated structures of ReAV mutants represent two crystallographic space groups. The monoclinic structures (R47A, S80A, K263A) contain two ReAV dimers in the asymmetric unit (ASU), while the orthorhombic structures (S48A, K51A) have only one ReAV dimer per ASU. The final resolution of the datasets is high, ranging from 1.40 to 1.95 Å (Table 3).

#### 3.4.1. The R47A Mutant

The crystal structure of this mutant revealed that the absence of Arg47 caused only minor structural rearrangements and atomic shifts in the active site region (Figure 2A), which were nearly identical in the four protein chains. The substitution of Arg47 by alanine affects the conformation of Glu17, which can no longer form H-bonds to the Nη atom of Arg47 of the complementary subunit. In protein chain A, the side chain of Glu17 is rotated toward the Nε atom of Arg12 from the same chain, while in chain C this residue forms H-bonds with the Oγ atom of Thr10 from the same chain and two water molecules. Interestingly, in protein chains B and D, Glu17 adopts two alternative conformations: one with one of its Oε atoms pointing to the side chain of Thr10 and another forming an H-bond with the Nε atom of Arg12 (Figure 2B). The metal binding site is conserved in the mutant structure and fully occupied with Zn^2+^. High difference electron density peaks were detected in the close vicinity of the Ser48 hydroxyl, as previously observed in both the ReAIV and ReAV crystal structures, and these peaks were interpreted as three water molecules forming very short H-bonds with the Ser48 Oγ atom (O…O 2.30–2.50 Å), as well as within their triangle (2.40–2.50 Å) (Figure 2C).

The hydrogen bond network at the Ser-Lys tandems (Ser48-Lys51 and Ser80-Lys263) closely resembles that of the WT protein [15] (Figure 1B and Figure 2A). The Oγ atom of Ser48 forms H-bonds to the Nζ atom of Lys51 and the Oγ atom of Ser80. The side chain of Lys51 is involved in hydrogen bonding with the Oγ atoms of Ser48 and Ser80, as well as with the carbonyl oxygen of Ala79. The Oγ atom of Ser80 from the second Ser-Lys tandem interacts with the Ser48 hydroxyl group, further stabilizing the protein structure [15]. The Nζ atom of Lys263 forms H-bonds with the Oγ atom of Ser80, carbonyl oxygen of Leu264, and a water molecule. Only one conformer of Asp187 is observed in the electron density maps, with one of its Oδ atoms forming an H-bond to the water molecule from the metal coordination sphere.

#### 3.4.2. The S48A Mutant

Mutating position 48 to alanine resulted in only minor atomic shifts and local conformational changes within the active site area, consistent in both protein chains. In the S48A mutant, the zinc cation is only partially present at the metal binding site, with a refined occupancy of 0.7, as is the water molecule from the zinc coordination sphere (Figure 3A). The loss of Ser48 from the Ser48-Lys51 tandem affects the conformational state of Lys51. The major rotamer of Lys51 (refined occupancy 0.7) retains its typical position where it forms H-bonds to the Oγ atom of Ser80, the carbonyl O atom of Ala79, and the Oδ atom of Asn134. The minor conformer, however, points in the opposite direction, forming an H-bond with a water molecule (Figure 3B). The Oγ atom of Ser80 no longer interacts with the Ser48 hydroxyl; instead, this interaction is compensated by an H-bond to an additional water molecule, which likely mimics a water molecule of the triad. The pattern of H-bonds at Lys263 remains unchanged compared to the WT protein. A negatively charged sulfate ion, derived from the crystallization solution, appears in the close vicinity of the metal binding site, forming H-bonds to the Nη atom of Arg47 and the main-chain N atoms of Gly188 and Cys189. Moreover, the SO_4_^2−^ anion is surrounded by five water molecules, one of which belongs to the metal coordination sphere (Figure 3C). Notably, the side chain of Asp187 adopts two conformations: one in which its Oδ atoms interact with the Oγ atom of Thr193 and the Nε atom of Arg47, and another in which the side chain is oriented toward the metal coordination sphere.

#### 3.4.3. The K51A Mutant

Substitution of Lys51 with alanine induced notable structural rearrangements within the active site region, and these changes are not strictly equivalent in the two protein chains. In chain A, the absence of Lys51 affects the position and conformational state of Lys138 in the metal coordination sphere (Figure 4A,B). The side chain of Lys138 adopts two alternative conformations: one rotamer moves away from the metal cation and forms H-bonds with the Oε atom of Gln54 and the Oγ atom of Ser48, while the other rotamer maintains its typical conformation, pointing toward the Zn^2+^ cation (Figure 4C). Furthermore, both the zinc cation and the coordinated water molecule are only partially present in the mutant crystal structure, with a refined occupancy of 0.5. The Ser48 side chain has a slightly different position depending on the protein chain, without the characteristic pattern of three closely spaced water molecules. In protein chain A, the side chain of Ser48 still forms an H-bond with the Oγ atom of Ser80, but the S48 residue is directed downward to the site previously occupied by the side chain of the mutated Lys51 residue (Figure 4B). In protein chain B, Ser48 retains its typical conformation, pointing toward the water molecule in the metal coordination sphere; however, the interaction with the Oγ atom of Ser80 is no longer possible. As observed in the previously described S48A mutant structure, a sulfate ion (at 0.5 occupancy) from the crystallization solution appears in the active site area, forming H-bonds with the Nη atom of Arg47, the main-chain N atoms of Gly188 and Cys189, and one or more water molecules, depending on the protein chain. Two conformers of Asp187 are visible in the electron density maps, and their occupancy can be correlated with the presence of the sulfate ion. One rotamer of Asp187 points to the main-chain N atoms of Cys189, Asn190, and Leu191, moving its anionic side chain away from the negatively charged SO_4_^2−^ ion. When there is no sulfate ion in the mutant structure, the Asp187 side chain is rotated toward the water molecule from the zinc coordination sphere, with which it forms an H-bond in addition to the salt-bridge interaction with the Nε and Nη atoms of Arg47. The pattern of H-bonds at the Ser80-Lys263 tandem is roughly the same as in the WT protein.

Structural analysis and careful inspection of the electron density maps revealed the presence of an unintended, second mutation at position 238, where His was expected. At the beginning of the refinement process, a strong positive Fo-Fc electron density peak was visible at the apex of the His238 ring (Figure 4D), in contrast to the clear His238 electron density calculated for the wild-type protein (PDB ID 7os5) (Figure 4E). An OMIT map calculated for residue 238 strongly suggested that a tyrosine was present at this position (Figure 4F). This spontaneous mutation most likely occurred during plasmid amplification after sequencing. Consequently, the amplified plasmid was subjected to additional sequencing, which revealed the presence of a second point mutation, specifically a C→U substitution in the histidine CAU codon, leading to the UAU codon coding for tyrosine. The His238 residue is poorly conserved in other bacterial orthologs of Class 3 L-asparaginases, and tyrosine is a possible natural substitution observed at this position. Therefore, it can be safely assumed that the H238Y mutation, which preserves the aromatic character, is distant from the active site, and is not implicated in any way in the catalytic mechanism, does not affect the protein’s activity. Thus, the crystal structure of the K51A mutant discussed in this work actually corresponds to the double K51A/H238Y mutant. To avoid confusion, we refer to this double mutant as the K51A variant.

#### 3.4.4. The S80A Mutant

The introduction of alanine at position 80 resulted in minor structural rearrangements of the protein in the active site area, which were nearly identical in the four protein chains. The S80A substitution affected the hydration pattern of Ser48, which is no longer surrounded by the triad of closely spaced water molecules (Figure 5A,B). The Oγ atom of Ser48 is H-bonded to the Nζ atom of Lys51 and two water molecules, resembling two water molecules of the typical “triangle”. Moreover, in protein chain A, the Ser48 residue adopts two conformations. One rotamer interacts with the two neighboring water molecules and the Nζ atom of Lys51, while the other, in addition to interacting with the side chain of Lys51, forms an H-bond with the Nζ atom of Lys263 from the second tandem (Figure 5B).

The Nζ atom of Lys263 is no longer able to interact with the Oγ atom of Ser80, but it continues to form H-bonds with the neighboring water molecule and the carbonyl oxygen of Leu264. The fully occupied Zn^2+^ binding site is conserved in the mutant structure. As observed in the S48A and K51A mutants, a sulfate ion from the crystallization solution is located at full occupancy near the metal binding site, forming H-bonds with the Nη atom of Arg47 and the main-chain N atoms of Gly188 and Cys189, as well as with several water molecules, one of which belongs to the metal coordination sphere. Only one conformer of Asp187 is observed in the electron density maps, with its Oδ atoms forming H-bonds to the Nε atom of Arg47, the Oγ atom of Thr193, and two water molecules. The side chain of Cys249 is most likely oxidized and was modeled as cysteinesulfinic acid (Figure 5C). Oxidative modification at the Sγ atom of Cys249 was observed in several ReAV structures reported by Loch et al. (2021) [15] and Pokrywka et al. (2024) [28]. The potential role of the Cys249 modification is not yet clear, but it has been shown that absence of this residue perturbs the pattern of H-bond connections in the active site area, disrupting the entire ReAV fold [28].

#### 3.4.5. The K263A Mutant

The absence of Lys263 led to only subtle structural rearrangements and atomic shifts within the active site, largely consistent in the four protein chains. A water molecule fills the available space left by the lysine side chain, marking the site of its Nζ atom and maintaining the original pattern of hydrogen bonds (Figure 6A,B). The Lys263 mutation affected the conformational state and hydration pattern of Ser48. The major rotamer of Ser48, with a refined occupancy of 0.7, can no longer form H-bonds with the Oγ atom of Ser80 and the Nζ atom of Lys51. Instead, it interacts with the carbonyl oxygen atom of Leu264 and two water molecules, mimicking the arrangement of two water molecules in the typical triangular configuration, though with increased O…O distances (2.70–2.90 Å).

The minor Ser48 conformer is oriented toward the metal binding site, forming H-bonds to the Oγ atom of Ser80, Nζ of Lys51, and three water molecules, one of which is coordinated to the metal cation. The zinc cation nearly fully occupies the metal binding site, with a refined occupancy of 0.8, shared with the coordinated water molecule. As observed in the previously described mutant structures, a sulfate ion (derived from the crystallization solution and present at 0.7 occupancy) is bound near the zinc cation, forming H-bonds with the main-chain N atoms of Gly188 and Cys189, the Nη atom of Arg47, and several water molecules, one of which is part of the metal coordination sphere (Figure 6C). Only one rotamer of Asp187 is observed in the mutant structure, forming H-bonds with the Nε atom of Arg47, Oγ atom of Thr193, and two water molecules. As observed in the S80A mutant structure, the side chain of Cys249 exhibits a chemical modification, modeled as S-hydroxycysteine.

### 3.5. Insights from the Structural and Functional Analysis of ReAV Mutants

Careful structural analysis, combined with kinetic data, provides insights into the importance of the residues selected for mutagenesis. The positively charged side chain of Arg47 supports the correct positioning of the L-Asn substrate molecule [34]. Moreover, Arg47 participates in a network of H-bonds stabilizing the protein structure by forming salt-bridge interactions with the carboxylate side chains of residues Glu17, Asp187, and Asp267 from the complementary subunit in the dimer. Thus, replacing the protonated Arg47 side chain with alanine disrupted the hydrogen bond network in this region, rendering the mutant protein inactive.

The role of the primary nucleophile of ReAV has been recently assigned to Ser48 [34]. In agreement with this postulate, substitution of Ser48 with alanine rendered the mutant protein inactive, as the L-asparaginase reaction could not be initiated. The Lys51 residue from the Ser48-Lys51 tandem, with its side chain unprotonated at the high pH (9.0) of the catalytic reaction, acts as a base for proton abstraction from Ser48 and participates in the proton shuttling network during the catalytic process [34]. The K51A mutein is incapable of L-asparagine hydrolysis because the Lys51 side chain can no longer play the role of the nucleophilic activator. The Lys138 residue from the metal coordination sphere changes its position in response to the K51A mutation as if to compensate for this absence, but ultimately this structural rearrangement is not able to “patch up” the disrupted catalytic apparatus. Another serine residue, namely Ser80 from the second Ser-Lys tandem, is an important hub for the H-bond network in the active site, interacting inter alia with the Ser48 nucleophile, to the point of forcing a non-planar configuration of the Ala79-Ser80 peptide bond [15]. Moreover, Ser80, together with Lys51 and Lys263, is involved in shuttling the proton abstracted from the Ser48 nucleophile [34]. The S80A mutation turned the protein activity off, as the proton shuttling network could not function properly. A similar situation was observed in the presence of the K263A mutation, where the absence of a lysine side chain could not be fully compensated by a water molecule. In the K263A mutant structure, the major rotamer of Ser48 shifts in response to the mutation, moving away from Lys51. This leads to stereochemistry that is suboptimal for the nucleophilic attack and prevents activation by the basic Lys51 side chain.

In some mutant structures (S48A, K51A, K263A), the zinc cation is modeled with fractional occupancy. These mutations affect residues in the vicinity of the metal cation and may weaken its binding, either directly or through altered coordination geometry. The mutations induce local conformational changes, such as alternative conformations of Lys138 (involved in zinc coordination), Lys51, or the Ser48 nucleophile.

## 4. Conclusions

The inducible Class 3 L-asparaginase ReAV exhibits no sequence or structural similarity to known enzymes from other classes of the L-asparaginase superfamily. However, it shares a similar fold with some glutaminases and serine β-lactamases. The recently described crystal structures of WT ReAV and its mutants in complex with the L-Asn ligand support the identification of Ser48 as the catalytic nucleophile [34]. To further investigate the catalytic mechanism of Class 3 L-asparaginases, we generated five ReAV mutants targeting highly conserved residues in the active site, namely Arg47, Ser48, Lys51, Ser80, and Lys263. Our mutagenesis studies revealed that substitution of any of these residues with alanine turned the catalytic activity off. The S48A mutant was unable to hydrolyze L-asparagine, as the reaction could not be initiated due to the absence of a nucleophile. Lys51, acting as a general base, is essential for activating Ser48. In the K51A mutein, the substituted residue is unable to fulfill this role, even with the positional shift of Lys138, which appears to compensate for the disrupted catalytic apparatus but ultimately fails to restore functionality. Ser80 plays a role not only in proton shuttling but also as a key hub in the hydrogen-bond network of the active site, interacting with the Ser48 nucleophile. This interaction contributes to a distortion of the Ala79-Ser80 peptide bond from planarity. The substitution of Ser80 with alanine abolished the enzymatic activity, likely due to disruption of the proton transfer network. Furthermore, the mutation led to protein aggregation, suggesting an additional role of Ser80 in maintaining the structural integrity of the protein. In the K263A mutant, the Ser48 residue is no longer properly activated by Lys51, as it moves away from Lys51 in response to the mutation. The R47A mutant was inactive as well, due to its inability to anchor the L-Asn substrate in the active site.

Together, our mutational and structural analyses demonstrate a tightly coordinated catalytic apparatus in ReAV, dependent on a highly conserved network of residues. These residues are individually essential for catalysis, substrate binding, and the maintenance of active-site integrity.

## Data Availability

Atomic coordinates and structure factors corresponding to the final crystallographic models of the ReAV mutants generated in this study have been deposited in the Protein Data Bank (PDB) under the accession codes: 9qct (R47A), 9qcu (S48A), 9qcw (K51A), 9qcy (S80A), and 9qcz (K263A). The corresponding raw diffraction images have been deposited in the Macromolecular Xtallography Raw Data Repository (https://mxrdr.icm.edu.pl) under DOI numbers: 10.60884/DZULXI (R47A), 10.60884/LKKYEO (S48A), 10.60884/KU8XZI (K51A), 10.60884/WKWVVZ (S80A), and 10.60884/NYTYTH (K263A).

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
