# Peer review of "Probing the Active Site of Class 3 L-Asparaginase by Mutagenesis: Mutations of the Ser-Lys Tandems of ReAV"

_biomolecules, 2025, doi:10.3390/biom15070944_

Round 1

Reviewer 1 Report

Comments and Suggestions for Authors

The authors analyze the role of five residues in the active site of the class 3 L-asparaginase ReA V, in particular of the residues of the Ser-Lys pairs. Analysis involves purification, single crystal X-ray diffraction structure determination, activity assays with Nessler's reagent and heat-rate shift experiments with ITC. 
Chemical and physical activity assays (Nessler and ITC) both state that all analyzed mutants are inactive. While this confirms the initial assumption that these residues are essential for the reaction, the complete inactivity does not allow any further differentiation of their role. A previous paper by the autors (Pokrywka, FEBS J., 2025, ref. 22) already proposed the double displacement mechanism of action for the amide hydrolysis. A more detailed mechanism than before is not developed. Specifically, the catalytic residues of the second displacement step could not be elucidated. 
The present paper is technically sound and well written, but lacks novelty to most of the readers but the most specialized. 

Specific comments: 

line 59: replace 
Class 1 L-asparaginases
by
Class 1 type II L-asparaginases

line 126
Since Ser48 has been identified as the nucleophile, I recommend including it in the scheme or mentioning it in the figure legend. 

line 244
Isn't the effect of higher enzyme concentration primarily an increase instead of an improvement of the signal?

Additional note: 
The set of three tightly bound water molecules at Ser48OG is an unusual and on first sight doubtable feature. I recommend further analysis of the hydrogen bonds and protonation states, though not in the current paper. The difference to a phosphate or borate ester (compare page 7 in Loch, Nature Comm., 2021) could be checked by XRD. 

Author Response

The authors analyze the role of five residues in the active site of the class 3 L-asparaginase ReA V, in particular of the residues of the Ser-Lys pairs. Analysis involves purification, single crystal X-ray diffraction structure determination, activity assays with Nessler's reagent and heat-rate shift experiments with ITC.

Chemical and physical activity assays (Nessler and ITC) both state that all analyzed mutants are inactive. While this confirms the initial assumption that these residues are essential for the reaction, the complete inactivity does not allow any further differentiation of their role. A previous paper by the autors (Pokrywka, FEBS J., 2025, ref. 22) already proposed the double displacement mechanism of action for the amide hydrolysis. A more detailed mechanism than before is not developed. Specifically, the catalytic residues of the second displacement step could not be elucidated.

Indeed, at this stage it is still not possible to fully fathom the details of the second step of the catalytic reaction.

The present paper is technically sound and well written, but lacks novelty to most of the readers but the most specialized.

With this work, we complete the mutagenic survey of the residues in and around the catalytic site of Class 3 L-asparaginase, and confirm the correctness of previous speculations.

Specific comments:

line 59: replace Class 1 L-asparaginases by Class 1 type II L-asparaginases

Done.

line 126

Since Ser48 has been identified as the nucleophile, I recommend including it in the scheme or mentioning it in the figure legend.

Ser48 is now mentioned in Scheme 1 caption.

line 244 Isn't the effect of higher enzyme concentration primarily an increase instead of an improvement of the signal?

Yes, we apologize for this oversimplification. Particularly in the case of a microcalorimeter, which has a very low noise level, we agree that it is more accurate to use the term 'increase' of a signal, and we have revised the text accordingly. For example, when using 2 µM of the R47A mutant, we observed a ΔDP of 0.085 µcal/sec in the kinetic MIM experiments (Fig. S1). Increasing the protein concentration to 10 µM resulted in a similar ΔDP of 0.079 µcal/sec, indicating that the signal did not scale with protein concentration. Both values remain below the recommended detection limit of 0.1 µcal/sec.

Additional note:

The set of three tightly bound water molecules at Ser48OG is an unusual and on first sight doubtable feature. I recommend further analysis of the hydrogen bonds and protonation states, though not in the current paper. The difference to a phosphate or borate ester (compare page 7 in Loch, Nature Comm., 2021) could be checked by XRD.

We thank the Reviewer for these suggestions. Experimental work on dissecting the “hydration pattern” of Ser48 and on differentiation between phosphate or borate as possible ester modifications is currently in progress.

Reviewer 2 Report

Comments and Suggestions for Authors

This manuscript investigates the structure and catalytic mechanisms of the recently characterized Class 3 L-asparaginase ReAV. It provides a comprehensive structural and mutational analysis to identify critical residues in the active site. The authors have conducted site-directed mutagenesis on five conserved residues (Arg47, Ser48, Lys51, Ser80, and Lys263), demonstrating their essential roles in catalysis through kinetic and crystallographic evidence. Notably, the work highlights Ser48 as the catalytic nucleophile, and elaborates on the functional interdependence among catalytic residues, contributing valuable insights into a structurally unique L-asparaginase. If authors revise the following points, I highly recommend the manuscript for publication in Biomolecules.

  1. While structural similarity to glutaminases and b-lactamases in briefly mentioned, the manuscript would benefit from a more in-depth discussion on the evolutionary implications of this fold conservation.

  1. Although the catalytic inactivity of the mutants is described, the absence of quantitative kinetic data for WT and mutants weakens the strength of some conclusions. Inclusion of such data, even as a table, would significantly improve the rigor of the analysis.

  1. The term “completely abolished” is used repeatedly but needs clarification. Was there truly zero measurable activity, or was it reduced below detection limits? Providing a threshold or limit of detection would clarify interpretation.

The manuscript is scientifically sound, well-organized, and addresses an important knowledge gap in the field of enzymology and structural biology. With the inclusion of quantitative kinetic data and deeper discussion on structural homology, it will make a strong contribution to the literature.

Author Response

Are the conclusions supported by the results? Can be improved.

We have now rewritten the conclusion section to more closely reflect the structural and kinetic results.

This manuscript investigates the structure and catalytic mechanisms of the recently characterized Class 3 L-asparaginase ReAV. It provides a comprehensive structural and mutational analysis to identify critical residues in the active site. The authors have conducted site-directed mutagenesis on five conserved residues (Arg47, Ser48, Lys51, Ser80, and Lys263), demonstrating their essential roles in catalysis through kinetic and crystallographic evidence. Notably, the work highlights Ser48 as the catalytic nucleophile, and elaborates on the functional interdependence among catalytic residues, contributing valuable insights into a structurally unique L-asparaginase. If authors revise the following points, I highly recommend the manuscript for publication in Biomolecules.

We thank the Reviewer for this encouraging summary of our work.

While structural similarity to glutaminases and b-lactamases in briefly mentioned, the manuscript would benefit from a more in-depth discussion on the evolutionary implications of this fold conservation.

The fold of Class 3 L-asparaginases is a versatile scaffold for several enzymatic activities, including b-lactam hydrolysis, although the acquisition of the metal cation and tight two-fold homodimerization are novel features acquired by L-asparaginases, or lost by the other structural homologs. This structural fold appears to have diversified through alteration in active-site architecture and substrate-binding regions, enabling adaptation to distinct substrates and catalytic functions across different enzyme families. It is also interesting to know that the genes of Class 3 L-asparaginases  have been spreading in the bacterial world not only by evolution but also by horizontal gene transfer (HGT). For example, pan-genomic analysis revealed the surprising fact that the ReAV gene was acquired by R. etli from Burkholderia, while the other isoform, ReAIV, originated in Rhizobia much earlier.

Although the catalytic inactivity of the mutants is described, the absence of quantitative kinetic data for WT and mutants weakens the strength of some conclusions. Inclusion of such data, even as a table, would significantly improve the rigor of the analysis.

We are grateful for this suggestion. We have now included the results from the Nessler assays and raw calorimetric data as Supplementary material to the manuscript.

The term “completely abolished” is used repeatedly but needs clarification. Was there truly zero measurable activity, or was it reduced below detection limits? Providing a threshold or limit of detection would clarify interpretation.

We thank the Referee for this comment. We agree with the Reviewer that some of the original wording was too definitive given the detection limits of the methods used, i.e. 0.01 U of asparaginase activity for the Nessler method (as reported for a 1 cm cuvette, whereas we used a 96-well plate with approximately one-third the optical path) and 0.1 μcal/s for heat change detection by ITC. As suggested, we have now included information on these detection limits in the manuscript along with appropriate references. Additionally, the corresponding results from both methods have been added as Supplementary material.

However, even if the methods theoretically had no detection limits, we believe that analyzing potential trace levels of asparaginase activity would be unjustified. This is because we are not aware of any reliable method to conclusively confirm or rule out trace contamination with WT plasmid or protein, which is handled in the same laboratory. Even minute contamination at low picomolar concentrations could generate a measurable signal.

The manuscript is scientifically sound, well-organized, and addresses an important knowledge gap in the field of enzymology and structural biology. With the inclusion of quantitative kinetic data and deeper discussion on structural homology, it will make a strong contribution to the literature.

We thank the Reviewer for this conclusion.

Reviewer 3 Report

Comments and Suggestions for Authors

The work by Pokrywka et al., titled “Probing the Active Site of Class 3 L-asparaginase by Mutagenesis. II. Mutations of the Ser-Lys Tandems of ReAV,” describes the crystal structures of five single-point mutants of the L-asparaginase ReAV from Rhizobium etli and compares these structures to that of the wild-type protein. These mutants are designed to confirm the catalytic roles of the amino acid residues forming the Ser-Lys tandem (Ser48, Lys51, Ser80, Lys263) and the Arg47 residue. None of these enzyme variants exhibited activity, as shown by activity assays.

As evidenced by previous studies from this research group on this enzyme, the experimental approach used here is robust. From this perspective, I see no criticism. Notably, the study provides five high-quality crystal structures, reflecting a significant amount of experimental work.

However, I consider this work to be a contribution of cumulative science: the results are not novel but instead serve to confirm previously known data from the same group. This is explicitly stated by the authors in the manuscript (see the conclusions section), where it is noted that Ser48 is the nucleophile, and Lys51, Ser80, and Lys263 participate in proton abstraction and shuttling. This study does not demonstrate these roles but rather confirms them.

The study primarily describes subtle structural rearrangements around the amino acid substitutions. Surprisingly, the authors include in the manuscript an “unintended” amino acid change—His238 replaced with Tyr. Unintended experimental issues should be avoided unless they have significant, unexpected functional or structural consequences.

Additional comments:

  • The first half of the abstract (lines 12-21) should be moved to the introduction. After providing a brief contextual background, only the results should be summarized in the abstract.
  • The panels in Figure 1 have been previously reported by this group in other publications (see Figure 1 in Pokrywka et al., Front. Chem. 12:1381032, doi: 10.3389/fchem.2024.1381032).
  • The same applies to Scheme 1.

Taken into account all these considerations (cumulative work, and lack of novelty), my recommendation is to reject the manuscript.

Author Response

Is the research design appropriate? Can be improved.

We are not quite sure how the research design, i.e. mutation - enzyme kinetic measurements - crystal structure determination and analysis, could be improved.

The work by Pokrywka et al., titled “Probing the Active Site of Class 3 L-asparaginase by Mutagenesis. II. Mutations of the Ser-Lys Tandems of ReAV,” describes the crystal structures of five single-point mutants of the L-asparaginase ReAV from Rhizobium etli and compares these structures to that of the wild-type protein. These mutants are designed to confirm the catalytic roles of the amino acid residues forming the Ser-Lys tandem (Ser48, Lys51, Ser80, Lys263) and the Arg47 residue. None of these enzyme variants exhibited activity, as shown by activity assays.

This is an elegant summary of our work.

As evidenced by previous studies from this research group on this enzyme, the experimental approach used here is robust. From this perspective, I see no criticism. Notably, the study provides five high-quality crystal structures, reflecting a significant amount of experimental work.

We thank the Reviewer for this positive assessment of the experimental aspect of our work.

However, I consider this work to be a contribution of cumulative science: the results are not novel but instead serve to confirm previously known data from the same group. This is explicitly stated by the authors in the manuscript (see the conclusions section), where it is noted that Ser48 is the nucleophile, and Lys51, Ser80, and Lys263 participate in proton abstraction and shuttling. This study does not demonstrate these roles but rather confirms them.

It is correct that our work has a confirmatory character. However, it is a necessary element of our project. So far the assessment of the essential catalytic role of the “S-K tandem” residues was hypothetical. We consider an experimental confirmation of this hypothesis to be necessary.

The study primarily describes subtle structural rearrangements around the amino acid substitutions. Surprisingly, the authors include in the manuscript an “unintended” amino acid change—His238 replaced with Tyr. Unintended experimental issues should be avoided unless they have significant, unexpected functional or structural consequences.

We think that we provided sufficient justification and description of corrective measures in the manuscript. As explained, the unintended mutation is in fact quite common in this Class of proteins, as the His238 residue is poorly conserved in other bacterial orthologs (~52%), and tyrosine is a possible natural substitution at this position.

Additional comments:

The first half of the abstract (lines 12-21) should be moved to the introduction. After providing a brief contextual background, only the results should be summarized in the abstract.

As recommended, we have substantially shortened the first part of the abstract. However, we feel that complete removal of lines 12-21 would compromise readability. Likewise, moving the incriminated lines to introduction is not a good idea because similar information is already there.

 The panels in Figure 1 have been previously reported by this group in other publications (see Figure 1 in Pokrywka et al., Front. Chem. 12:1381032, doi: 10.3389/fchem.2024.1381032). The same applies to Scheme 1.

We see nothing wrong in using a similar (not at all identical) figure to illustrate the main points of our paper. Without Fig. 1 and its panels the discussion presented in the paper would be quite difficult to follow. The same applies to Scheme 1. However, we have used this remark as a prompt to slightly modify Fig. 1 (panel C) for better visibility.

Taken into account all these considerations (cumulative work, and lack of novelty), my recommendation is to reject the manuscript.

We are sorry to see this verdict and, obviously, do not agree with it.

Round 2

Reviewer 3 Report

Comments and Suggestions for Authors

The authors have partially addressed the concerns I raised in the initial version of the manuscript.

Regarding the "unintended" amino acid substitution (H238Y), the authors should clarify whether this change has any biological significance. Based on the manuscript, it appears that it is an experimental artifact. If this is the case, it is not scientifically relevant and should be removed from the manuscript (text and figure).

It is strongly recommended to prepare figures that do not resemble previously published ones. Self-plagiarism also affect figures.

Author Response

The authors have partially addressed the concerns I raised in the initial version of the manuscript.

We thought we addressed all the concerns, although we elected to stand by our own version/presentation in some cases.

Regarding the "unintended" amino acid substitution (H238Y), the authors should clarify whether this change has any biological significance. Based on the manuscript, it appears that it is an experimental artifact. If this is the case, it is not scientifically relevant and should be removed from the manuscript (text and figure).

There was already a statement in the manuscript (P14) about the lack of biological relevance of the H238Y spontaneous mutation in the K51A variant, but we have somewhat rephrased it now to make it stronger. Respectfully, we take issue with the recommendation that the H238Y mutation should be “removed from the manuscript”. It is a physical observation and has to be reported, rather than swept under the carpet. Besides, we stress (again) the important aspect of this discovery, namely that this mutation was discovered at the stage of electron density map interpretation, thus pointedly illustrating the power of X-ray crystallographic structure determination, allowing one to correct the amino acid identity in protein sequence several hundred residues long! A short note emphasizing this aspect has been added on P10.

It is strongly recommended to prepare figures that do not resemble previously published ones. Self-plagiarism also affect figures.

This remark is rather strange as it seems to suggest that use of the same artistic style and indeed graphics program to illustrate different structural aspects is culpable! We see nothing wrong with the superficial semblance of the present illustrations to some previous ones (and indeed, similarity among Figures 2-6 of the present manuscript). The bottom line is that each of those figures illustrates a different structural feature.